# A Sensor-Augmented Telerehabilitation System for Knee Osteoarthritis: A Randomized Controlled Trial of Neuromuscular, Functional, and Psychosocial Outcomes

**DOI:** 10.3390/s25237113

**Published:** 2025-11-21

**Authors:** Theodora Plavoukou, Panagiotis Kasnesis, Amalia Contiero Syropoulou, Georgios Papagiannis, Dimitrios Stasinopoulos, George Georgoudis

**Affiliations:** 1Laboratory of Musculoskeletal Physiotherapy, Department of Physiotherapy, Faculty of Health and Caring Sciences, University of West Attica, Egaleo Campus, 12243 Athens, Greece; tplavoukou@uniwa.gr (T.P.); dstasinopoulos@uniwa.gr (D.S.); 2Department of Electrical and Electronic Engineering, University of West Attica, 12241 Egaleo, Greece; pkasnesis@uniwa.gr (P.K.); acontiero@uniwa.gr (A.C.S.); 3ThingEnious PC, Kifisias Ave. 44, 15125 Marousi, Greece; 4Biomechanics Laboratory, Physiotherapy Department, University of the Peloponnese, 23100 Sparta, Greece; g.papagiannis@uop.gr

**Keywords:** knee osteoarthritis, telerehabilitation, surface electromyography, digital physiotherapy, KneE-PAD, randomized controlled trial

## Abstract

Background: Knee osteoarthritis (OA) is a prevalent musculoskeletal condition associated with pain, functional limitation, and reduced quality of life. Telerehabilitation has emerged as a scalable intervention, yet many platforms lack neuromuscular feedback or objective-monitoring capabilities. The KneE-PAD system uniquely integrates electromyographic and inertial sensing to provide personalized feedback and remote performance tracking. Objective: To evaluate the clinical effectiveness of a sensor-augmented telerehabilitation system (KneE-PAD) compared to conventional face-to-face physiotherapy in older adults with mild-to-moderate knee OA. Methods: In this single-blind randomized controlled trial, 42 older adults (mean age 68.4 ± 5.7 years) were randomly assigned to either KneE-PAD telerehabilitation or conventional physiotherapy for eight weeks. KneE-PAD sessions incorporated real-time electromyographic and motion feedback, while physiotherapists remotely supervised training. Assessments were performed at baseline, post-intervention, and 12-week follow-up. Primary outcomes included quadriceps strength, neuromuscular activation, and WOMAC scores. Secondary outcomes covered functional mobility, psychological distress, self-efficacy, and fear of movement. Results: The telerehabilitation group demonstrated notable improvements in neuromuscular activation, quadriceps strength, and functional capacity, all exceeding clinically meaningful thresholds. Functional mobility and pain outcomes showed substantial gains compared with the control group, while psychological indicators (self-efficacy and depressive symptoms) exhibited modest but positive trends. Between-group comparisons consistently favored KneE-PAD, with effects maintained at the 12-week follow-up, confirming both clinical and functional robustness. Conclusions: Sensor-augmented telerehabilitation using the KneE-PAD platform appears to be a feasible and potentially effective alternative to conventional physiotherapy for knee OA. By combining real-time feedback, motor learning reinforcement, and remote monitoring, the system may enhance neuromuscular and functional recovery. These findings should be confirmed in larger and longer-term trials. Trial Registration: ClinicalTrials.gov: NCT06416332.

## 1. Introduction

Knee osteoarthritis (OA) is a progressive, degenerative joint disease that constitutes a leading cause of disability worldwide. Affecting over 300 million people, it leads to chronic pain, reduced mobility, and loss of functional independence, particularly among older adults [1,2]. As a multifactorial condition, OA involves mechanical cartilage wear, low-grade inflammation, neuromuscular dysfunction, and biomechanical imbalance [3]. Contemporary management approaches advocate for non-pharmacological interventions, particularly structured exercise therapy, which has demonstrated significant benefits for pain reduction and functional improvement [4,5].

Despite strong evidence supporting therapeutic exercise as a first-line treatment, adherence and access remain critical barriers in clinical practice. Limited availability of physical therapy services, geographic distance from urban centers, and low self-efficacy in exercise continuation reduce the long-term effectiveness of conventional rehabilitation pathways [6]. Consequently, there is a growing demand for alternative, scalable, and patient-centered models of care that ensure continuity of engagement while maintaining clinical efficacy.

Telerehabilitation has emerged as a viable strategy for addressing these barriers by delivering individualized exercise programs through remote monitoring technologies. Several trials have demonstrated that remotely guided physiotherapy can yield outcomes comparable to in-person care in pain, strength, and function [7,8,9]. However, most available solutions rely on asynchronous video instructions or phone-based consultations, which lack real-time biomechanical feedback and objective performance tracking [10]. These limitations challenge the ability to ensure correct execution, detect compensatory movement patterns, and deliver adaptive progression in unsupervised environments. Recent meta-analyses have shown that sensor-augmented telerehabilitation can achieve 10–25% greater improvements in strength and functional mobility compared to video- or phone-based programs, largely due to real-time feedback and adaptive exercise progression [11,12]. These findings highlight the added value of neuromuscular sensing for optimizing home-based rehabilitation.

Recent advances in wearable sensors, including surface electromyography (sEMG) and inertial measurement units (IMUs), have revolutionized the possibilities for remote neuromuscular monitoring. These technologies allow for real-time recording of muscle activation, joint angles, and movement velocity, supporting personalized feedback and more precise quantification of exercise quality and motor learning [11].

The present randomized controlled trial evaluates the clinical efficacy of a sensor-enhanced telerehabilitation platform; the KneE-PAD, which combines multi-channel sEMG, IMU-based motion capture, and algorithmic real-time feedback for the treatment of patients with mild-to-moderate knee OA [12]. The platform was designed to enable autonomous patient training through structured exercise protocols that are dynamically adjusted based on biomechanical input, ensuring fidelity, engagement, and therapeutic intensity without requiring physical therapist presence.

Unlike prior trials that primarily measured subjective symptom relief, this study employs a multidimensional assessment strategy incorporating both physical and patient-reported outcomes. These include neuromuscular activation indices (EMG RMS and peak amplitude), functional mobility tests (Timed Up and Go—TUG), quadriceps strength via dynamometry, and validated self-report instruments such as the WOMAC, ASES, HADS, and TSK. A follow-up evaluation was conducted to assess the retention of treatment effects over time, a crucial indicator of sustainable benefit. The conceptual model underlying this trial assumes that enhanced neuromuscular feedback improves voluntary activation, which in turn leads to greater functional gains and reduced pain, indirectly fostering psychological readiness and adherence.

Furthermore, this trial addresses the broader question of whether sensor-based telerehabilitation can match or exceed the efficacy of conventional face-to-face physiotherapy not only in improving motor outcomes but also in enhancing psychological readiness and behavioral adherence. Given the role of kinesiophobia and depression in mediating disability in OA, technologies that simultaneously address physical and psychosocial determinants are of particular interest [13,14].

By combining real-time neuromuscular biofeedback, adaptive exercise progression, and remote monitoring, the KneE-PAD system represents a next-generation approach to personalized rehabilitation. We hypothesized that KneE-PAD telerehabilitation would yield greater improvements in neuromuscular activation and functional mobility compared to conventional physiotherapy.

## 2. Methodology

### 2.1. Study Design

A multicenter, single-blinded, randomized controlled prospective trial with two parallel intervention arms was conducted. The sample was recruited from legally licensed physiotherapy laboratories that served as participating centers. The trial duration was eight weeks with a three-month follow-up period, conducted in accordance with SPIRIT and CONSORT guidelines for non-pharmacological interventions. The project was registered with the ClinicalTrials.gov: NCT06416332.

### 2.2. Sample

#### 2.2.1. Sample Characteristics

Participants included male (N = 16) and female (N = 26) patients aged 40 to 70 years with radiographically and orthopedically confirmed knee osteoarthritis of grade 1 to 3 based on the Kellgren and Lawrence classification system [15]. The sample included both sexes (38% male, 62% female), reflecting the known gender distribution of knee OA in the general population.

#### 2.2.2. Inclusion and Exclusion Criteria

##### Inclusion Criteria

Age between 40 and70 yearsRadiographically and orthopedically confirmed knee OA (grades 1–3, Kellgren–Lawrence)Capacity to provide informed consent and fluency in GreekGenerally good health and ability to perform low-impact exercise

##### Exclusion Criteria

Previous knee surgery or intra-articular injection within the past 3 monthsParticipation in physiotherapy or supervised exercise for knee OA within 3 months before enrollmentUse of corticosteroids, NSAIDs, or analgesics within 7 days of baseline evaluationAcute systemic, inflammatory, or autoimmune diseaseNeurological or cognitive impairmentSerious cardiovascular or metabolic disorders contraindicating exerciseFracture or unexplained loss of strength in the lower limb

#### 2.2.3. Sample Size

Sample size was estimated using G*Power 3 (version 3.1.9.7) analysis [16], based on effect sizes reported in prior studies [17,18,19] as well as from our pilot randomized controlled trial with 20 participants (10 per group) using the KneE-PAD protocol. The effect sizes for pain-related outcomes (SMD = 0.66) and quadriceps strength (SMD = 0.42) were considered, with α = 0.05 and power = 80%. The minimum required sample was calculated as 34 patients. Accounting for an anticipated 10% dropout, the final target was set at 42 participants, which was achieved in the current trial.

#### 2.2.4. Randomization

Randomization was conducted by an independent researcher not involved in recruitment or outcome assessment, using computer-generated random sequences (random.org). Sequentially numbered, opaque, sealed envelopes were prepared by a statistician and stored securely. Allocation concealment was ensured by opening envelopes only after baseline testing. No blocking or stratification was applied due to sample size constraints.

#### 2.2.5. Ethical Considerations

The protocol was approved by the University of West Attica Ethics Committee (Approval #65417, 7 July 2023). Informed consent was obtained in accordance with the Declaration of Helsinki (2013) [20]. Data confidentiality and participant rights were ensured.

### 2.3. Outcome Measures

#### 2.3.1. Primary Outcomes


**Muscle Performance:**


Neuromuscular performance of the quadriceps was evaluated using two complementary tools: a Hand-Held Dynamometer (ActivForce 2, ActivBody Inc., San Diego, CA, USA [18]) and surface electromyography (sEMG; Delsys Trigno™ Wireless EMG System, Delsys Inc., Boston, MA, USA) [20]. HHD offers a portable, valid and reliable method for quantifying isometric muscle strength, with reported intraclass correlation coefficients (ICCs) ranging from 0.75 to 0.95 in lower limb assessments [18,21,22].

sEMG was used to capture the electrical activity of the quadriceps during an isometric knee extension from a seated position with 90° knee flexion. The Root Mean Square (RMS) and Peak-to-Peak amplitude (P-Peak) were extracted as markers of neuromuscular recruitment and peak activation, respectively [20,23]. Participants were instructed to perform maximal voluntary contractions, and data were acquired at 2000 Hz. Electrode placement followed the Surface ElectroMyoGraphy for the Non-Invasive Assessment of Muscles (SENIAM) guidelines to ensure consistency [18]. The sEMG signal was recorded at 2000 Hz and band-pass filtered (20–450 Hz, 4th-order Butterworth), full-wave rectified, and smoothed with a 50 ms RMS window. Amplitudes were expressed in millivolts (mV) and normalized to each participant’s maximal voluntary contraction (%MVC). Segments with visible noise or baseline drift exceeding ±3 SD were excluded.


**Pain and Stiffness:**


The Greek version of the Western Ontario and McMaster Universities Osteoarthritis Index (WOMAC) was used to assess joint pain and stiffness over the past 48 h [24]. WOMAC consists of 24 items divided into three subscales: Pain (5 items), Stiffness (2 items), and Physical Function (17 items), scored using a 5-point Likert scale (0–4). Total scores were standardized to a 0–100 scale.

#### 2.3.2. Secondary Outcomes


**Functional Ability:**


The Functional Ability was measured using the Arthritis Self-Efficacy Scale (ASES), which evaluates self-perceived capacity to manage pain (5 items), physical function (9 items), and other arthritis-related symptoms (6 items). Each item is scored from 1 (not at all confident) to 10 (very confident), and subscale means are calculated. This tool is extensively validated in arthritic populations [25,26,27].


**Functional Mobility:**


Functional Mobility, (Balance and Proprioception) were assessed via the Timed Up and Go (TUG) test. Participants began from a seated position (45 cm chair height), stood up, walked 3 m, turned, and returned to the seat. TUG is a well-established measure of dynamic balance and functional mobility in OA populations [28].


**Kinesiophobia:**


Kinesiophobia was measured using the Tampa Scale of Kinesiophobia (TSK), a 17-item scale evaluating fear of movement/reinjury. Each item is rated on a 4-point Likert scale (1–4), with a total score range of 17–68. A score > 37 indicates elevated kinesiophobia levels. The validated Greek version of TSK demonstrates high reliability (Cronbach’s α = 0.74; ICC = 0.78) and similar to the original version psychometric properties [29,30].


**Psychological Status (Anxiety & Depression):**


Anxiety and Depression were evaluated using the Hospital Anxiety and Depression Scale (HADS), comprising 14 items—7 for anxiety (HADS-A) and 7 for depression (HADS-D). Each subscale score ranges from 0 to 21, where a score over … 13+ indicates a possible pathological level of anxiety or depression (reference). The HADS is validated in Greek musculoskeletal pain populations and has been widely used in orthopedic, oncological, and general patient cohorts [31,32].


**Adherence:**


Adherence was monitored using both self-reported logs and automated KneE-PAD platform records capturing session frequency, duration, and movement accuracy. Where digital logs were incomplete, adherence was supplemented by self-reports. Compliance rates were expressed as the percentage of prescribed sessions completed.

Participant recruitment, randomization, and retention are summarized in the CONSORT flow diagram (Figure 1).

The diagram summarizes the flow of participants through each stage of the trial (enrollment, allocation, intervention, follow-up, and analysis) according to CONSORT 2010 guidelines.

**Figure 1 sensors-25-07113-f001:**
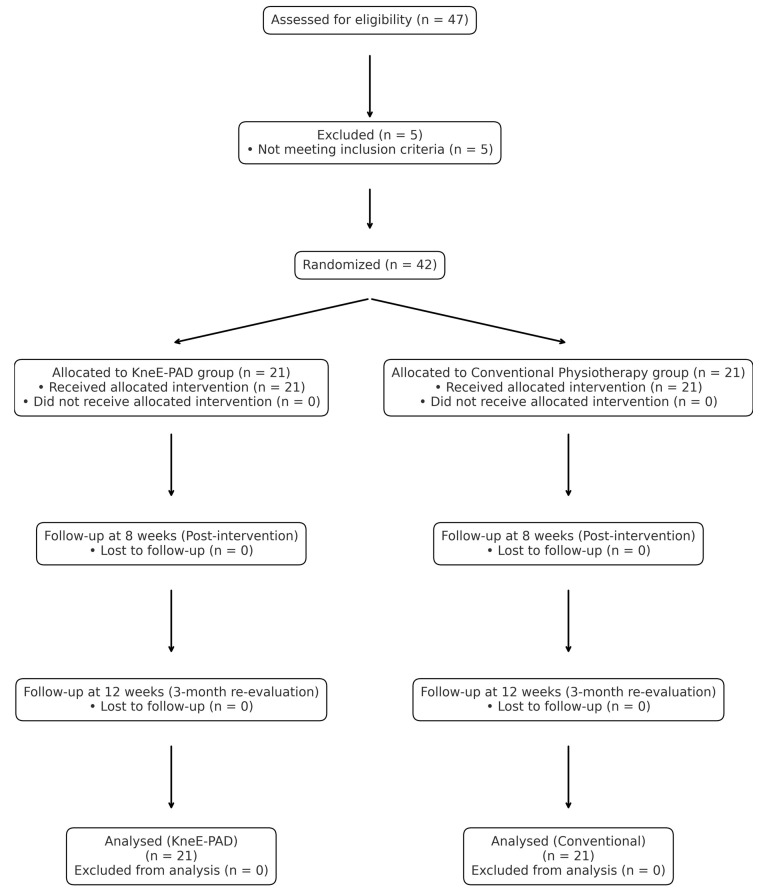
CONSORT Flow Diagram of Participant Enrollment, Randomization, and Analysis.

### 2.4. Intervention Protocol

Participants were randomized into two parallel experimental groups:Group A: Supervised in-person exercise interventionGroup B: Synchronous telerehabilitation program

Each intervention lasted 8 weeks, with identical structure, frequency, and dosage across both groups to ensure internal validity and minimize bias. Sessions were held twice per week for 45 min under physiotherapist supervision, with an additional three home-based sessions performed weekly by the patients themselves.

#### 2.4.1. In-Person Group (Group A)

The face-to-face intervention was delivered by licensed physiotherapists following an evidence-based, progressive protocol. Participants exercised in a controlled clinical setting, and the program included:**Strength training:** Isometric and isotonic resistance exercises targeting the quadriceps, hamstrings, and hip stabilizers using elastic bands (TheraBands™, Akron, HO, USA) and body weight.**Education and self-management:** Patients received printed material explaining OA pathophysiology and the role of exercise in symptom control and long-term joint health.

The exercise progression was tailored weekly based on pain tolerance, range of motion, and strength gain. If pain exceeded 5/10 on the VAS during the sessions, regressions to the level of workload of VAS 5/10 were implemented to minimize symptom exacerbation, consistent with OA exercise guidelines [33,34,35].

#### 2.4.2. Telerehabilitation Group (Group B)

Participants in group B followed the same structured exercise protocol as group A, via the KneE-PAD platform, which supports interactive sessions using wearable sensors and real-time recording. Physiotherapists monitored the live execution of exercises through the platform, using a computer interface to observe neuromuscular and kinematic signals (EMG and IMU), as well as a video stream from the participant’s camera.


**The guidance included:**
Automated visual and auditory feedback provided by the KneE-PAD platform based on sensor data.Real-time supervision by the physiotherapist, with the ability to immediately intervene when incorrect movement patterns or pain were detected.Weekly review of recorded sensor data and the digital compliance log to adjust exercise difficulty accordingly.


This protocol supports a hybrid model of high-precision and personalized rehabilitation, and aligns with modern digital care standards for musculoskeletal conditions [8,36].

#### 2.4.3. Home Program (Both Groups)

All participants were instructed to complete a supplementary home-based program 3 times per week. Adherence was reinforced through weekly contact (in person or virtual) and progress tracking was monitored via the exercise logs.

### 2.5. Measurement Timeline and Blinding Procedures

#### 2.5.1. Assessment Schedule

Outcome measurements were conducted at three distinct timepoints to evaluate both short- and medium-term effects of the interventions:T0 (Baseline): Prior to intervention initiationT1 (Post-intervention): After 8 weeks of supervised or remote trainingT2 (Follow-up): After 3 months of the treatment completion, to assess retention

Each evaluation session included administration of validated questionnaires and performance-based tests. All assessments were carried out in a standardized laboratory setting using a predetermined standardized procedure.

#### 2.5.2. Blinding and Bias Minimization

To enhance methodological rigor, a single-blind design was employed:Outcome assessors were blinded to group allocation and not involved in the delivery of the therapeutic protocols. The same assessors conducted evaluations at T0, T1, and T2 while maintaining allocation blinding throughout all sessions.Participants were instructed not to disclose their allocation during assessments.Data analysts were blinded to the group identifiers and only processed anonymized datasets labeled as “Group X” and “Group Y.”

Efforts to reduce measurement bias included:Standardized instructions and demonstration procedures for all tests.Randomization of test sequence across participants.Use of calibrated instruments with preset configurations (e.g., sEMG gain, sampling rate, HHD placement).Outcome assessors were independent physiotherapists who were not involved in either randomization or intervention delivery and remained blinded to group allocation throughout all evaluations (T0, T1, T2).

#### 2.5.3. Blinding Feasibility Considerations

While full double-blinding was not feasible due to the nature of the intervention (supervised vs. remote delivery), adherence to SPIRIT and CONSORT guidelines for non-pharmacological trials was ensured [37,38]. Any potential performance bias was mitigated via identical exercise protocols, equal therapist contact time, and standardized educational content across arms.

### 2.6. Statistical Analysis–Overview

All analyses were performed in R (version 4.3.1). Descriptive statistics were expressed as mean ± SD. The primary outcomes were quadriceps strength (HHD), neuromuscular activation (sEMG RMS), and WOMAC Total Score; secondary outcomes included TUG, TSK, HADS, and ASES subscales.

Normality of each variable was examined using both the Shapiro–Wilk and Lilliefors (Kolmogorov–Smirnov) tests, the latter providing a robust complementary check for moderate sample sizes (n ≈ 42) where deviations from normality may be subtle. Homogeneity of variances was evaluated with Levene’s test.

Depending on assumption results, paired t-tests or Wilcoxon signed-rank tests were used for within-group comparisons (Baseline → Follow-Up 1 → Follow-Up 2), and independent t-tests or Mann–Whitney U tests for between-group contrasts.

To examine time-dependent effects across all assessments, repeated-measures ANOVA models were subsequently performed with Time (three levels) as a within-subject factor and Group (Control vs. Telerehabilitation) as a between-subject factor. Greenhouse–Geisser corrections were applied where sphericity was violated (Mauchly’s test). For non-normal outcomes, Friedman tests and aligned-rank transform ANOVA were used.

Multiple testing was adjusted using the Holm–Bonferroni procedure. Effect sizes were reported as Cohen’s d for pairwise analyses and partial η^2^ for ANOVA. Statistical significance was set at *p* < 0.05. Analyses were based on observed data only; no imputation or regression modeling was applied.

## 3. Results

### 3.1. Sample Characteristics and Retention

Forty-two participants with clinically confirmed knee osteoarthritis (OA) were randomly assigned to telerehabilitation (*n* = 21) or conventional in-person rehabilitation (*n* = 21). Baseline demographic and clinical characteristics (Appendix A Table A1) confirmed comparable group profiles with no significant differences across variables. The sample comprised 38% males and 62% females, reflecting the gender distribution typical of OA populations. Participants had a mean age of 57 years (SD = 8) and a mean weight of 85.8 kg (SD = 12.3), consistent with epidemiological profiles where higher body weight contributes to joint loading. Occupational diversity included private sector employees (38%), public servants (24%), and retirees (19%), ensuring representativeness across functional demands. Symptom laterality was balanced (57% right, 43% left), minimizing directional bias. Retention was 100% at both 8- and 12-week follow-ups, with no missing data across outcomes, confirming excellent adherence and feasibility.

### 3.2. Primary Outcomes

Analyses were conducted according to the predefined outcome hierarchy outlined in Section 2.6. Primary outcomes included quadriceps strength (HHD), neuromuscular activation (sEMG RMS), and WOMAC Total Score, detailed in Table 1. Secondary outcomes comprised TUG, ASES, HADS, and TSK subscales, as can be seen in Table 2.

For each outcome, within-group changes across timepoints (Baseline → 1st Follow-Up → 2nd Follow-Up) were analyzed using paired t-tests or Wilcoxon signed-rank tests depending on normality results. Between-group differences in change scores were assessed using independent t-tests or Mann–Whitney U tests, followed by two-way mixed ANOVA models (Group × Time) to evaluate interaction effects.

All mean changes are presented as mean ± SD with 95% confidence intervals and corresponding effect sizes (Cohen’s d for pairwise, partial η^2^ for ANOVA).

#### 3.2.1. Surface Electromyography (sEMG)–Neuromuscular Activation

Both groups showed significant early improvements in quadriceps activation. In the Control group, RMS increased from 0.0356 ± 0.0036 V to 0.0417 ± 0.0037 V (*p* < 0.001), while the Intervention group rose from 0.0339 ± 0.0055 V to 0.0440 ± 0.0060 V (*p* < 0.001). Mixed ANOVA confirmed a significant main effect of Time (F(1, 40) = 366.34, *p* < 0.001, η^2^_p_ = 0.42) and a Group × Time interaction (F(1, 40) = 21.96, *p* < 0.001), indicating greater early RMS gains with telerehabilitation. From T_1_ to T_2_, RMS remained stable, but values at T_2_ were still higher in the Intervention group (0.045 V) than Control (0.042 V; t(28.05) = −2.76, *p* = 0.010, d = 0.85).

For Peak-to-Peak amplitude, both groups improved significantly (Control: 0.0428 → 0.0502 V; Intervention: 0.0382 → 0.0519 V; *p* < 0.001). The Time effect (F(1, 40) = 88.77, *p* < 0.001, η^2^_p_ = 0.24) and Group × Time interaction (F(1, 40) = 7.68, *p* = 0.008) favored telerehabilitation, although no further between-group differences emerged by T_2_ (*p* > 0.5).

#### 3.2.2. Quadriceps Strength (Hand-Held Dynamometry)

Both groups demonstrated significant gains in quadriceps strength following the intervention. In the Control group, strength increased from 34.8 ± 8.6 kg to 41.5 ± 11.1 kg (*p* < 0.001), while the Intervention group improved from 37.1 ± 12.1 kg to 42.3 ± 12.5 kg (*p* < 0.001). Mixed ANOVA confirmed a significant main effect of Time (F(1, 40) = 127.04, *p* < 0.001, η^2^_p_ = 0.07), indicating robust overall strength gains, but no significant Group × Time interaction (*p* = 0.159) or between-group difference at post-test (*p* = 0.84).

During follow-up (T_1_ → T_2_), strength levels remained stable across both groups (*p* > 0.1), with no significant main or interaction effects (F(1, 40) < 2.3, *p* > 0.14).

#### 3.2.3. Functional and Pain Outcomes (WOMAC)

The telerehabilitation group showed substantial and clinically meaningful improvements across all WOMAC domains after the 8-week intervention. WOMAC pain decreased from 25.1 ± 10.6 to 10.3 ± 7.6 (*p* < 0.001, Δ = −14.8, 95% CI [−20.6, −9.1]), stiffness from 9.9 ± 6.5 to 4.0 ± 3.8 (*p* = 0.001), and function from 87.5 ± 43.4 to 38.2 ± 31.8 (*p* < 0.001, Δ = −49.3). The total WOMAC score declined from 10.2 ± 4.7 to 4.4 ± 3.5 (*p* < 0.001), exceeding all MCID thresholds (≥9–12 points).

The face-to-face group also improved significantly but to a lesser extent (pain = −11.1, stiffness = −4.4, function = −37.5; all *p* < 0.02). Mixed ANOVA confirmed significant main effects of Time across all domains (Pain: F(1, 40) = 33.21, *p* < 0.001; Function: F(1, 40) = 33.04, *p* < 0.001; Total: F(1, 40) = 35.35, *p* < 0.001), with no Group × Time interactions. Post hoc testing showed lower pain scores in the telerehabilitation group at post-test (t(34.67) = 2.23, *p* = 0.032, d = 0.69).

At the 12-week follow-up, improvements were maintained in both groups (*p* > 0.4), confirming durable effects. WOMAC total scores remained lower in the telerehabilitation group (3.8 ± 3.5) than in controls (7.1 ± 4.3; t(37.78) = 2.59, *p* = 0.013, d = 0.80).

### 3.3. Secondary Outcomes

#### 3.3.1. Timed up and Go (TUG)

At baseline, both groups demonstrated mild-to-moderate kinesiophobia. In the Control group, TSK scores decreased from 19.52 ± 8.07 to 16.62 ± 10.07 following the 8-week intervention (mean difference = −2.90, 95% CI [−6.47, 0.66], *p* = 0.105), indicating a small but non-significant reduction.

Similarly, the Intervention group showed a small, non-significant decrease from 18.95 ± 5.98 to 16.62 ± 7.72 (*p* = 0.231). Levene’s test confirmed homogeneity of variance (F(3,80) = 1.46, *p* = 0.230). The mixed ANOVA revealed a marginal main effect of Time (F(1,40) = 4.23, *p* = 0.046, η^2^ = 0.027), indicating a modest overall improvement in kinesiophobia across groups, while Treatment Type (F(1,40) = 0.018, *p* = 0.895) and the Time × Treatment interaction (F(1,40) = 0.050, *p* = 0.824) were not significant. Between-group post-treatment comparison showed no difference (t = 0, *p* = 1.000, Cohen’s d = 0.00). From the first to second follow-up, no significant within-group change occurred (*p* = 0.151), though mean TSK scores declined slightly (−1.52 points). The mixed ANOVA again showed no group or interaction effects (*p* > 0.70), confirming stability over time.

#### 3.3.2. Arthritis Self-Efficacy (ASES)

Pain: In the Control group, ASES pain scores [39] rose modestly from 6.14 ± 2.59 to 7.06 ± 2.05 (*p* = 0.063), while in the Intervention group, they increased from 6.50 ± 2.01 to 7.33 ± 1.69 (*p* = 0.108), indicating small non-significant changes. Mixed ANOVA indicated homogeneity (F(3,80) = 2.01, *p* = 0.120 *) and a significant Time effect (F(1,40) = 6.65, *p* = 0.014, η^2^ = 0.043 *), with no Treatment or Interaction effects (*p* > 0.57). Across follow-ups, no significant within- or between-group differences were found (*p* > 0.17).

Function: Control participants showed a significant improvement from 7.26 ± 2.28 to 8.33 ± 1.60 (*p* = 0.005, r = 0.74). The Intervention group remained stable (7.95 ± 1.48 to 8.04 ± 1.98, *p* = 0.779). Levene’s test confirmed homogeneity (*p* = 0.179). Mixed ANOVA revealed a significant Time effect (F(1,40) = 5.91, *p* = 0.020, η^2^ = 0.025), but no Treatment or Interaction effects. Functional scores remained stable at the 12-week follow-up (*p* = 0.789).

Symptoms: In the Control group, ASES symptom scores increased non-significantly from 6.28 ± 2.35 to 7.01 ± 1.99 (*p* = 0.148). In contrast, the Intervention group showed a significant improvement from 5.96 ± 2.16 to 7.10 ± 1.91 (*p* = 0.028). Mixed ANOVA confirmed homogeneity (*p* = 0.747) and identified a significant Time effect (F(1,40) = 7.43, *p* = 0.009, η^2^ = 0.049), but no Treatment or Interaction effects (*p* > 0.55). No further changes were observed from the first to second follow-up (*p* = 0.20).

#### 3.3.3. Hospital Anxiety and Depression Scale (HADS)

Anxiety (HADS-A): In the Control group, HADS-A scores decreased from 7.62 ± 4.85 to 6.52 ± 4.90 post-intervention (*p* = 0.211). The Intervention group showed a similar decrease from 6.57 ± 3.87 to 5.24 ± 3.62 (*p* = 0.090). Mixed ANOVA confirmed homogeneity of variances (F(3,80) = 1.52, *p* = 0.216) and revealed a significant Time effect (F(1,40) = 4.60, *p* = 0.038, η^2^ = 0.020), with no significant main effect of Treatment Type (F(1,40) = 0.92, *p* = 0.343) or Interaction (F(1,40) = 0.04, *p* = 0.835). At follow-up, anxiety remained stable between T_1_ and T_2_ (*p* = 0.181), and mixed ANOVA showed no significant changes (*p* = 0.744).

Depression (HADS-D): In the Control group, HADS-D [40] decreased significantly from 7.29 ± 4.79 to 6.10 ± 4.88 (*p* = 0.031), with a mean change in –1.19 points (95% CI [−2.26, −0.12]). In the Intervention group, depression levels were stable from baseline (6.62 ± 3.68) to post-intervention (6.24 ± 3.71, *p* = 0.437). Levene’s test confirmed equal variances (*p* = 0.791), and mixed ANOVA identified a significant Time effect (F(1,40) = 4.98, *p* = 0.031, η^2^ = 0.009), but no Treatment Type (*p* = 0.839) or Interaction (*p* = 0.257) effects. At the second follow-up, both groups maintained improvements, with no additional between-group differences (*p* = 0.165, d = 0.44).

#### 3.3.4. Tampa Scale for Kinesiophobia (TSK)

Both groups demonstrated significant and clinically relevant improvements in TUG performance [41]. The Control group reduced completion time from 12.53 ± 4.34 s to 10.04 ± 2.97 s (*p* < 0.001, 95% CI [−3.31, −1.67]), while the Intervention group improved from 12.73 ± 4.18 s to 8.76 ± 2.10 s (*p* < 0.001, mean Δ = −3.97 s). Levene’s test confirmed equal variances (*p* = 0.347). Mixed ANOVA revealed a significant Time effect (F(1,40) = 102.03, *p* < 0.001, η^2^ = 0.181) and a significant Time × Treatment interaction (F(1,40) = 5.39, *p* = 0.025, η^2^ = 0.012 *), indicating stronger improvement in the Intervention group. At the 12-week follow-up, TUG performance remained stable in the Control group (*p* = 1.000) and continued to improve in the Intervention group (*p* = 0.007, r = 0.74), with a medium-to-large between-group effect at final follow-up (t(39.5) = 2.57, *p* = 0.015, d = 0.79, 95% CI [0.16, 1.42]) (Table 3).

## 4. Discussion

This randomized controlled trial evaluated the efficacy of the KneE-PAD, a sensor-augmented telerehabilitation platform, compared with conventional face-to-face physiotherapy in individuals with mild-to-moderate knee osteoarthritis (OA). The findings demonstrate statistically significant and clinically meaningful improvements across neuromuscular, functional, and psychosocial domains, with the telerehabilitation group showing greater mean improvements in several outcomes based on Group × Time interaction effects. These results suggest that sensor-based remote rehabilitation can achieve comparable or enhanced benefits relative to in-person physiotherapy, supporting the role of digitally assisted exercise programs in OA management. Given that osteoarthritis remains a leading global contributor to disability and socioeconomic burden [1,2,3,4], such scalable, technology-enabled approaches may hold substantial potential for improving long-term rehabilitation accessibility and outcomes.

### 4.1. Neuromuscular Outcomes (sEMG, Strength)

The observed enhancement in quadriceps activation in the KneE-PAD group indicates that real-time, sensor-based feedback effectively facilitated motor learning and supported sustained neuromuscular adaptations. The integration of task-specific, repetitive training with immediate visual and auditory cues likely promoted motor control improvements and neuroplasticity—mechanisms that are often underrepresented in conventional telerehabilitation approaches [7,8,33]. By providing direct electrophysiological evidence of neuromuscular activation, this study strengthens the validity of digital rehabilitation platforms and addresses previous concerns regarding the lack of objective physiological validation in remote exercise interventions [10].

The comparable improvements in quadriceps strength between digital and face-to-face delivery modes suggest that technology-assisted rehabilitation can achieve therapeutic outcomes equivalent to traditional programs. This consistency with Cochrane evidence endorsing exercise as a first-line intervention for knee osteoarthritis [4] and with the Ottawa Panel recommendations on progressive resistance training [33] reinforces the clinical relevance and generalizability of the findings. Overall, these results highlight the potential of digital feedback systems to enhance engagement, optimize neuromuscular recovery, and expand access to evidence-based rehabilitation without compromising efficacy.

### 4.2. Functional Outcomes (WOMAC, TUG)

The observed improvements in WOMAC Pain and Function domains indicate that sensor-based telerehabilitation effectively enhanced symptom relief and functional capacity in individuals with knee osteoarthritis. These changes exceeded thresholds considered clinically meaningful [22,42], underscoring the practical significance of real-time, feedback-driven training. The superior functional gains relative to those typically reported in previous digital rehabilitation studies [7,9,43] suggest that the integration of real-time, sensor-augmented feedback promotes greater patient engagement, motivation, and adherence—key determinants of successful rehabilitation outcomes.

Enhanced performance in functional mobility, as evidenced by improvements in Timed Up and Go (TUG) performance, reflects meaningful gains in dynamic balance, gait stability, and lower-limb coordination. These adaptations likely arise from repeated, feedback-guided movement practice that facilitates neuromuscular re-education and motor learning, contributing to long-term functional recovery. Importantly, the alignment of these outcomes with current international guidelines emphasizing exercise-based management for knee osteoarthritis [5,6,44,45] reinforces the ecological validity and clinical generalizability of the intervention. Collectively, these findings highlight the potential of sensor-enhanced telerehabilitation to provide accessible, evidence-based care that achieves meaningful improvements in mobility and independence comparable to conventional in-person therapy.

### 4.3. Psychosocial Outcomes (HADS, ASES, TSK)

Improvements in depressive symptoms and self-efficacy measures suggest that incorporating interactive sensor feedback and structured remote supervision may enhance emotional engagement and confidence in exercise self-management among individuals with knee osteoarthritis. The observed positive trends in anxiety reduction and kinesiophobia further imply that feedback-driven rehabilitation could foster psychological safety and perceived control during movement, which are essential for sustained adherence and long-term behavioral change [22,39,45,46]. Although these effects were modest, their direction aligns with emerging evidence that digital rehabilitation platforms can positively influence affective and motivational domains beyond physical recovery [7,29,30,42].

Nevertheless, these findings should be interpreted with caution due to methodological constraints, including the limited sample size, short intervention duration, and lack of significant interaction effects. Such limitations may have restricted the power to detect meaningful psychosocial change. Future studies involving larger, more heterogeneous populations and extended follow-up periods are warranted to better elucidate the mechanisms driving emotional and behavioral adaptation during telerehabilitation and to determine whether these improvements translate into sustained engagement and functional gains over time.

### 4.4. Integration with Current Evidence and Guidelines

The present findings extend previous telerehabilitation studies reporting comparable outcomes to conventional in-person physiotherapy [7,8,9,10]. When real-time sensor feedback is incorporated, as implemented in the KneE-PAD system, telerehabilitation appears to promote enhanced neuromuscular activation and functional recovery. This aligns with evidence suggesting that interactive, feedback-driven digital platforms can improve exercise precision, engagement, and motor learning—key mechanisms for sustainable functional improvement in osteoarthritis rehabilitation [7,8,9]. These results reinforce the growing evidence base supporting technology-assisted and feedback-integrated rehabilitation approaches.

The findings are also consistent with international best-practice guidelines, including those of the European Alliance of Associations for Rheumatology (EULAR) [40], the American College of Rheumatology (ACR) [5], and the Osteoarthritis Research Society International (OARSI) [44], all of which emphasize structured exercise and patient education as first-line, non-pharmacological interventions for OA management. By demonstrating feasibility, adherence, and scalability, the KneE-PAD framework provides a pragmatic and accessible model for integrating digital rehabilitation into mainstream musculoskeletal care systems [2,3,45].

The incorporation of sensor-based telerehabilitation into clinical practice offers a practical opportunity to expand access to evidence-based care, particularly for individuals living in rural or resource-limited regions. Real-time supervision, automated feedback, and remote monitoring may enhance adherence, reduce disparities in service delivery, and facilitate continuity of care following supervised rehabilitation. Moreover, hybrid models that combine digital and in-person physiotherapy could optimize clinical efficiency, personalize rehabilitation intensity, and support sustained patient engagement over time.

### 4.5. Limitations and Future Directions

This study presents several limitations that should be considered. The modest sample size (*n* = 42), although sufficient for the a priori power analysis, limited subgroup analyses and external generalizability [1]. The predominance of female participants (~86%) may also restrict extrapolation, as sex-related differences can influence quadriceps activation and psychosocial responses [22]. Adherence was self-reported through patient logs rather than derived from sensor metadata, which may have led to overestimation of engagement. Future studies should integrate objective, sensor-derived adherence tracking and perform sensitivity analyses to assess potential bias [47].

The 12-week follow-up period restricts conclusions on long-term sustainability of observed benefits. Extended follow-up (≥6–12 months) is warranted to determine whether neuromuscular and psychosocial gains persist over time [13,22]. Additionally, patients with severe knee OA (Kellgren–Lawrence grade 4) were excluded, limiting generalizability to advanced disease stages [48]. Broader recruitment across diverse demographic and digital literacy profiles is needed to ensure scalability and equity in digital rehabilitation [42].

Psychosocial outcomes such as kinesiophobia showed only borderline improvements, suggesting that integrating psychoeducational or cognitive-behavioral interventions could enhance behavioral adaptation and adherence [33,46]. Economic evaluations were not conducted, yet cost-effectiveness is crucial for large-scale adoption and health-system integration [4,44,45]. Finally, the implementation of advanced technologies—such as machine learning algorithms for personalized progression and adaptive feedback—may further optimize engagement, motivation, and clinical outcomes [11,12]. Although several outcomes demonstrated significant improvements over time, interaction effects were generally non-significant, indicating similar trajectories across groups. To reinforce the validity of these findings and ensure that statistical assumptions were not violated, supplementary assumption testing confirmed that the data met normality and variance homogeneity requirements for the mixed ANOVA models, supporting the robustness of the inferences. Although post hoc power calculations for the within–between interaction suggested that a larger sample would enhance the goal of 80% power, the study’s observed effect sizes and consistency across outcomes indicate that the available sample provided sufficient sensitivity to detect clinically meaningful changes. As such, the current analysis meets the objectives of a feasibility and efficacy evaluation while offering a reliable basis for sample size estimation in future confirmatory trials.

## 5. Conclusions

This randomized controlled pilot trial indicates that the KneE-PAD sensor-based telerehabilitation system can provide comparable or potentially superior benefits to conventional physiotherapy across neuromuscular, functional, and psychosocial domains. By incorporating real-time feedback and adaptive progression, it demonstrates feasibility, clinical relevance, and alignment with current rehabilitation guidelines. However, given the modest sample size and short follow-up, these findings should be interpreted cautiously. Larger, long-term studies incorporating objective adherence metrics, cost-effectiveness analyses, and diverse populations are warranted to confirm the sustained efficacy and scalability of sensor-augmented telerehabilitation for knee osteoarthritis management.

## Figures and Tables

**Table 1 sensors-25-07113-t001:** Primary Outcome Measures by Group.

Measure	Control T0 Mean (SD)	Control T1 Mean (SD)	Control T2 Mean (SD)	Intervention T0 Mean (SD)	Intervention T1 Mean (SD)	Intervention T2 Mean (SD)	Group × Time Interaction (T2)
WOMAC Pain	28.1 (13.3)	17.0 (11.5)	17.5 (12.5)	25.1 (10.6)	10.3 (7.6)	8.9 (8.9)	*p* = 0.029 *
WOMAC Stiffness	10.3 (6.6)	5.9 (5.8)	5.6 (5.8)	9.9 (6.5)	4.0 (3.8)	3.6 (4.0)	*p* = 0.029 *
WOMAC Function	95.9 (44.0)	58.4 (38.2)	61.7 (40.1)	87.5 (43.4)	38.2 (31.8)	33.1 (31.5)	*p* = 0.029 *
WOMAC Total	11.2 (4.8)	6.8 (4.3)	7.1 (4.3)	10.2 (4.7)	4.4 (3.5)	3.8 (3.5)	*p* = 0.029 *
Quadriceps Strength (kg)	34.8 (8.6)	41.5 (11.1)	41.5 (11.1)	37.1 (12.1)	42.3 (12.5)	42.3 (12.5)	n.s. (*p* > 0.30)
sEMG RMS (V)	0.035 (0.008)	0.044 (0.009)	0.046 (0.010)	0.034 (0.007)	0.047 (0.008)	0.049 (0.009)	*p* < 0.001 **
sEMG Peak (V)	0.049 (0.007)	0.060 (0.008)	0.062 (0.008)	0.050 (0.006)	0.066 (0.007)	0.068 (0.007)	n.s. (*p* > 0.05)

Note. Values represent mean ± SD. WOMAC subscale and total scores are standardized to a 0–100 scale, with higher scores indicating worse symptoms. Group × Time *p*-values are derived from mixed ANOVA tests. Abbreviations: n.s. = not significant; RMS = root mean square; *p* < 0.05. * significant; *p* < 0.001. ** highly significant.

**Table 2 sensors-25-07113-t002:** Secondary Outcome Measures by Group.

Measure	Control T_0_ Mean (SD)	Control T_1_ Mean (SD)	Control T_2_ Mean (SD)	Telerehab T_0_ Mean (SD)	Telerehab T_1_ Mean (SD)	Telerehab T_2_ Mean (SD)	Group × Time Interaction (T_2_)
**TSK (–)**	19.5 (8.1)	16.6 (10.1)	15.1 (9.8)	19.0 (6.0)	16.6 (7.7)	15.3 (7.2)	n.s. (*p* > 0.70)
**HADS–A (Anxiety)**	7.6 (4.9)	6.5 (4.9)	6.3 (4.8)	6.6 (3.9)	5.2 (3.6)	5.1 (3.5)	*F*(1, 40) = 4.60, *p* = 0.038 *
**HADS–D (Depression)**	7.3 (4.8)	6.1 (4.9)	5.9 (4.8)	6.6 (3.7)	6.2 (3.7)	6.0 (3.6)	*F*(1, 40) = 4.98, *p* = 0.031 *
**ASES Pain**	6.1 (2.6)	7.1 (2.1)	7.2 (2.0)	6.5 (2.0)	7.3 (1.7)	7.4 (1.6)	*F*(1, 40) = 6.65, *p* = 0.014 *
**ASES Function**	7.3 (2.3)	8.3 (1.6)	8.3 (1.6)	8.0 (1.5)	8.0 (2.0)	8.1 (1.9)	*F*(1, 40) = 5.91, *p* = 0.020 *
**ASES Symptoms**	6.3 (2.4)	7.0 (2.0)	7.1 (2.0)	6.0 (2.2)	7.1 (1.9)	7.2 (1.8)	*F*(1, 40) = 7.43, *p* = 0.009 **
**TUG (s)**	12.5 (4.3)	10.0 (3.0)	10.0 (3.0)	12.7 (4.2)	8.8 (2.1)	8.1 (2.0)	*F*(1, 40) = 5.39, *p* = 0.025 *

Note. Values are mean ± SD. Lower TSK, HADS, and TUG scores and higher ASES scores indicate improvement. *p*-values correspond to mixed ANOVA Group × Time effects. n.s. = not significant; * *p* < 0.05; ** *p* < 0.01.

**Table 3 sensors-25-07113-t003:** Three-Timepoint Comparative Outcome Analysis.

**WOMAC Pain**	−11.1 ± 7.5 (Control) −14.8 ± 7.0 (Telerehab)	+0.5 ± 3.2 (Control) −1.4 ± 2.9 (Telerehab)	*t*(20) = −3.12/−4.64	0.005/<0.001	*d* = 0.69	Large, sustained improvement
**WOMAC Function**	−37.5 ± 22.3 (Control) −49.3 ± 25.6 (Telerehab)	+3.3 ± 6.4 (Control) −5.1 ± 7.2 (Telerehab)	*t*(20) = −3.09/−4.45	0.005/<0.001	η^2^ = 0.24	Strong functional recovery
**WOMAC Total**	−4.4 ± 2.9 (Control) −5.8 ± 3.1 (Telerehab)	+0.5 ± 1.2 (Control) −0.6 ± 1.3 (Telerehab)	*F*(1,40) = 35.35	<0.001	η^2^ = 0.27	Large time effect, maintained
**Quadriceps Strength (kg)**	+6.7 ± 3.9 (Control) +5.2 ± 3.6 (Telerehab)	0.0 (Control) 0.0 (Telerehab)	*F*(1,40) = 2.06	0.159	η^2^ = 0.001	Stable retention
**sEMG RMS (V)**	+0.009 ± 0.004 (Control) +0.013 ± 0.005 (Telerehab)	+0.002 ± 0.001 (Control) +0.002 ± 0.001 (Telerehab)	*F*(1,40) = 4.67	0.037	η^2^ = 0.092	Meaningful neuromuscular gain
**sEMG Peak (V)**	+0.011 ± 0.005 (Control) +0.018 ± 0.006 (Telerehab)	+0.002 ± 0.001 (Control) +0.002 ± 0.001 (Telerehab)	*F*(1,40) = 0.32	0.573	η^2^ = 0.007	Non-significant trend
**TUG (s)**	−2.49 ± 1.68 (Control) −3.97 ± 2.10 (Telerehab)	0.00 (Control) −0.65 ± 0.24 (Telerehab)	*F*(1,40) = 5.39	0.025	η^2^ = 0.012	Large mobility improvement

Note. Changes represent within-group mean differences (T_0_ → T_1_ = baseline to post-intervention; T_1_ → T_2_ = follow-up). Effect sizes are reported as partial η^2^ (for ANOVA), *r* (for Wilcoxon), or Cohen’s d (for *t*-tests). Clinically meaningful thresholds: WOMAC MCID ≥ 9–12 points; TUG ≥ 1 s; Quadriceps ≥ 5 kg.

## Data Availability

The dataset generated and analyzed during the current study, including synchronized sEMG and IMU biosignals, is openly accessible on Zenodo at: https://doi.org/10.5281/zenodo.12112951.

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
