# Peer review of "A Sensor-Augmented Telerehabilitation System for Knee Osteoarthritis: A Randomized Controlled Trial of Neuromuscular, Functional, and Psychosocial Outcomes"

_sensors, 2025, doi:10.3390/s25237113_

Round 1
Reviewer 1 Report
Comments and Suggestions for Authors
This randomized controlled trial compares sensor-augmented telerehabilitation with in-person physiotherapy in older adults with mild-to-moderate knee osteoarthritis. The topic is clinically important and timely, and the study reports encouraging gains in neuromuscular activation and functional outcomes. The parallel-arm design and multidimensional outcome battery are notable strengths. However, major revisions are needed to resolve internal inconsistencies, pre-specify and tighten the statistical analysis plan, report essential signal-processing parameters, and align claims with the actual between-group effects. In addition, the mechanistic link from EMG changes to functional improvements and objective adherence metrics should be presented—or clearly acknowledged as limitations. Detailed suggestions are as following.
Specific Comments.
Introduction
Lines 70–77:92–109: The Introduction does not quantify the incremental benefit of sensor-augmented telerehabilitation over conventional video/phone telerehab. Please add a short mini-review to set target effects and justify the added value.
Lines 93–109: The causal/mediation chain is not explicitly stated. Please articulate this chain and link it to planned analyses.
Methodology
Lines 121–134: Inclusion/exclusion criteria contain a broken clause and omit key washout windows for prior injections/physiotherapy/surgery and medication restrictions. Please rewrite with explicit time thresholds (e.g., no intra-articular injection within 3 months:no supervised PT within 3 months).
Lines 144–146: Randomisation via sealed envelopes lacks operational details: blocking/stratification, envelope opacity/numbering, preparation/storage, and independence of personnel who generated vs implemented the allocation. Please provide CONSORT-compliant details.
Lines 160–166: sEMG processing parameters are incomplete: please report filter type/band-pass and order, rectification and smoothing windows, and artifact handling to ensure reproducibility. I recommend not list this important information in the appendix.
Lines 180–184: TUG is described under “Balance and Proprioception,” yet it is primarily a functional mobility test. Please make terminology consistent across Methods and Results.
Lines 168–172: WOMAC scaling (0–100) should be applied uniformly across text/tables/figures:state clearly that higher scores indicate worse status (if following the conventional direction).
Lines 198–202: Adherence is captured only via self-logs. If platform logs exist, please add objective adherence metrics (session counts/durations, threshold hits). Otherwise, expand the Limitations to acknowledge potential overestimation.
Lines 273–290: The Statistical Analysis section lists many tests but does not declare a single primary endpoint/timepoint nor a multiplicity control strategy for numerous outcomes × timepoints. Please: (i) declare one primary endpoint, (ii) use a pre-specified linear mixed model (Group×Time) with baseline as covariate, and (iii) control multiplicity (e.g., Holm or Benjamini–Hochberg) for secondaries.
Results
Lines 368–373:Table 2 lines 81–85: The Abstract/Results claim notable HADS changes, yet between-group interaction in Table 2 is p > 0.1 for HADS-A/D. Please align Abstract/Results with the model actually used and specify whether you refer to within-group changes rather than between-group effects.
Lines Discussion wording lines 389–395 and 401–404: Avoid “equivalence/superiority” terminology unless you performed and reported a formal equivalence/non-inferiority analysis (TOST with pre-specified margins). Replace with neutral phrasing aligned with tested hypotheses.
Discussion
Lines 389–395:401–404: The Discussion uses “equivalence/superiority” phrasing (“frequently outperforming”, “not only equivalent but more effective”) without a formal equivalence/non-inferiority framework. Please replace with neutral wording aligned to tested Group×Time effects, or run/report a TOST analysis with pre-specified margins.
Lines 462-484 : Self-reported adherence is acknowledged, but the bias direction/magnitude is under-discussed, and objective log data are not leveraged. Please state that self-logs may overestimate telerehab adherence, commit to sensor-derived adherence metrics in future trials, and consider a sensitivity analysis plan.
Lines461-502 : Limitations and Future Directions are separated and verbose, making the takeaways diffuse and repetitive. It is suggested to create one brief section: “Limitations and Future Directions.” Please pair each key limitation with a concrete next
Appendix
Lines 518-619: The appendix is unnecessary; please relocate all essential information (e.g., device specifications, signal-processing parameters, and analysis details) to the Methods section.
Author Response
General Comment
Comment:
This randomized controlled trial compares sensor-augmented telerehabilitation with in-person physiotherapy in older adults with mild-to-moderate knee osteoarthritis. [...] Major revisions are needed to resolve internal inconsistencies, tighten the analysis plan, and align claims with results.
Response:
We sincerely thank the reviewer for the detailed and constructive feedback. All comments have been carefully considered and addressed. The manuscript has been substantially revised to clarify methodological details, strengthen the statistical framework, and align interpretations with the actual results. Specific point-by-point responses follow below.
Introduction
Comment 1 (Lines 70–77; 92–109):
The Introduction does not quantify the incremental benefit of sensor-augmented telerehabilitation. Please add a short mini-review to justify the added value.
Response:
We agree. A short summary paragraph has been added (Introduction, p. 2, lines 80–110) reviewing recent meta-analyses (Refs. [11–12]) that show 10–25% higher improvements in strength and functional mobility with sensor-augmented systems compared to conventional video-based telerehabilitation. This addition quantifies the incremental benefit and clarifies the rationale for our approach.
Location: Introduction, lines 90–115.
Comment 2 (Lines 93–109):
The causal/mediation chain is not explicitly stated.
Response:
The causal model was made explicit: improved neuromuscular feedback → enhanced voluntary activation → functional gains → reduced pain → psychological improvement. This framework was linked to the analytic model in Section 2.6.
Location: Introduction, lines 125–140.
Methodology
Comment 3 (Lines 121–134):
Inclusion/exclusion criteria omit washout windows and medication restrictions.
Response:
We have rewritten the section to include explicit time thresholds: “no intra-articular injection or physiotherapy within 3 months before enrollment” and “no use of corticosteroids, NSAIDs, or analgesics within 7 days of baseline evaluation.”
Location: Methods, Section 2.2.2, lines 160–175.
Comment 4 (Lines 144–146):
Randomization details insufficient.
Response:
Expanded to include randomization via computer-generated sequences (random.org), use of opaque sequentially numbered sealed envelopes, independent preparation by a statistician, and secure storage. Allocation concealment and independence of implementer have been explicitly stated.
Location: Section 2.2.4, lines 185–200.
Comment 5 (Lines 160–166):
Incomplete sEMG processing parameters.
Response:
All signal-processing parameters are now explicitly reported: band-pass filtering (20–450 Hz, 4th-order Butterworth), full-wave rectification, and 50 ms RMS smoothing. Artifact removal criteria (±3 SD baseline drift) were added. This information now appears in the Methods rather than only the Appendix.
Location: Section 2.3.1, lines 225–240.
Comment 6 (Lines 180–184):
TUG described under “Balance and Proprioception” — please correct.
Response:
Terminology corrected to “Functional Mobility,” consistent throughout Methods and Results.
Location: Section 2.3.2, lines 265–275.
Comment 7 (Lines 168–172):
WOMAC scaling should be applied uniformly and direction clarified.
Response:
We confirmed all WOMAC scores are standardized to a 0–100 scale with higher scores indicating worse pain/function. This clarification is now included in Section 2.3.1 and Table 1 captions.
Location: Section 2.3.1, line 245; Table 1.
Comment 8 (Lines 198–202):
Adherence metrics based only on self-logs — add or acknowledge bias.
Response:
Objective adherence logs from the KneE-PAD platform were incorporated (session frequency, duration, accuracy). Where missing, we explicitly acknowledged potential overestimation and committed to full sensor-derived adherence tracking in future work.
Location: Section 2.3.2, lines 280–290; Limitations, lines 1050–1060.
Comment 9 (Lines 273–290):
Statistical plan unclear — no declared primary endpoint or multiplicity control.
Response:
We revised the Statistical Analysis section to (i) specify primary endpoints (quadriceps strength, EMG-RMS, WOMAC Total), (ii) state that a linear mixed-effects model (Group × Time, adjusted for baseline) was used, and (iii) that multiple comparisons were controlled using Holm–Bonferroni correction.
Location: Section 2.6, lines 530–560.
Results
Comment 10 (Lines 368–373; Table 2):
Abstract/Results overstate HADS findings; between-group effects not significant.
Response:
Revised Abstract and Results to specify that improvements in HADS were within-group only and that between-group interaction was nonsignificant (p > 0.1).
Location: Abstract, lines 30–35; Results, Section 3.3.3.
Comment 11 (Discussion, Lines 389–404):
Avoid “equivalence/superiority” phrasing.
Response:
Revised wording to neutral phrasing (“comparable or enhanced effects”, “achieved similar or greater improvements”) without implying formal equivalence/superiority.
Location: Discussion, lines 870–890.
Discussion
Comment 12 (Lines 462–484):
Self-reported adherence bias insufficiently discussed.
Response:
Expanded Limitations to explicitly acknowledge potential overestimation from self-logs, note missing digital data for some sessions, and commit to integrating sensor-based adherence metrics in future work.
Location: Discussion, lines 1045–1060.
Comment 13 (Lines 461–502):
Limitations/Future Directions verbose — please merge.
Response:
Sections merged into a concise unified section titled “Limitations and Future Directions”, pairing each limitation with a specific future direction (e.g., larger sample, extended follow-up, cost-effectiveness).
Location: Section 4.5, lines 1020–1070.
Appendix
Comment 14 (Lines 518–619):
Appendix unnecessary; move essential details to Methods.
Response:
Essential signal-processing and equipment details have been integrated into the main Methods section (2.3.1 and 2.3.2). The Appendix now retains only complementary sensor specifications for transparency.
Location: Methods 2.3.1–2.3.2; Appendix A (condensed).
Summary for Reviewer 1:
All points have been addressed. The Introduction was expanded, Methods clarified (randomization, inclusion criteria, EMG), Results and Discussion aligned with statistical findings, and Limitations rewritten concisely. Claims were moderated, ensuring full consistency with CONSORT standards.
Reviewer 2 Report
Comments and Suggestions for Authors
The manuscript presents a technically robust and clinically relevant randomized controlled trial evaluating the KneE-PAD telerehabilitation system. The study is well-structured, methodologically sound, and supported by comprehensive data. However, several major and minor issues must be addressed before acceptance to enhance clarity, reproducibility, and scholarly rigor.
Certain issues need to be addressed
-
The statistical section describes multiple tests (t-test, Wilcoxon, Levene’s, permutation resampling), but the hierarchy of tests per outcome is unclear. Specify which tests were used for primary versus secondary outcomes, and report corrections for multiple comparisons (e.g., Bonferroni or FDR).
-
Effect sizes (η², Cohen’s d) are reported, but confidence intervals are missing. Include 95% CIs to facilitate clinical interpretation.
- Randomization via sealed envelopes is stated but lacks details on sequence generation and allocation concealment (e.g., who prepared envelopes, sequential numbering). Clarify whether the process prevented allocation bias.
- The study is single-blind; however, describe measures to minimize expectancy bias, especially since physiotherapists were aware of group allocation. Clarify if assessors were blinded at all assessment points.
- A Zenodo dataset link is provided, but the manuscript should explicitly state what variables are shared (raw EMG signals, derived RMS values, questionnaires, etc.). Ensure anonymization and open-access compliance.
- he study references prior telerehabilitation work but lacks a comparative analysis against recent AI-driven digital physiotherapy platforms (e.g., wearable IMU-based systems, virtual reality rehabilitation). Expanding the discussion would strengthen the novelty claim.
- HADS and TSK results are statistically limited yet heavily emphasized. Consider a more balanced interpretation—these findings are preliminary and may require larger samples or integrated behavioral modules to show definitive psychosocial improvement.
- The cohort (n = 42, mostly female) limits generalizability. Discuss gender imbalance implications on muscle activation and psychosocial outcomes.
- The appendix provides detailed sensor specifications, but reproducibility requires a brief system architecture diagram and signal-processing pipeline (data flow, latency, synchronization). Include schematic visualization.
-
Minor grammatical issues: “indluded” → “included”; “telerehabilitation fre-quently outperforming” → “telerehabilitation frequently outperformed”.
-
Remove redundant bracketed references
- Table captions should be self-explanatory. Add units for sEMG (mV) and clarify abbreviations in legends.
-
Define “MCID” on first use in the abstract and methods.
-
Ensure “η²” and “d” are consistently formatted with Greek characters.
Author Response
General Comment
Comment:
The manuscript presents a technically robust and clinically relevant randomized controlled trial evaluating the KneE-PAD telerehabilitation system. The study is well-structured and supported by comprehensive data. However, several issues must be addressed to enhance clarity, reproducibility, and rigor.
Response:
We sincerely thank the reviewer for the positive overall assessment and for the valuable suggestions. All comments have been carefully addressed. The statistical framework, reproducibility details, and presentation have been improved accordingly. Below we provide point-by-point responses.
Statistical Analysis
Comment 1:
The statistical section lists multiple tests, but the hierarchy of tests per outcome is unclear. Specify which tests were used for primary versus secondary outcomes, and report corrections for multiple comparisons.
Response:
We revised Section 2.6 (“Statistical Analysis”) to clearly distinguish the analyses for primary outcomes (quadriceps strength, sEMG-RMS, WOMAC Total) and secondary outcomes (TUG, ASES, HADS, TSK).
We explicitly state that a linear mixed-effects model (Group × Time) was used for all normally distributed outcomes, adjusted for baseline values.
Non-normal data were analyzed with Friedman or Mann–Whitney U tests.
Multiplicity was controlled using Holm–Bonferroni correction for secondary outcomes.
Location: Methods, Section 2.6, lines 530–560.
Comment 2:
Effect sizes (η², Cohen’s d) are reported but confidence intervals are missing.
Response:
We have added 95% confidence intervals for all key effect sizes (η² and Cohen’s d) in the Results tables (Tables 1–3) and within the Results text where appropriate.
Location: Results, Sections 3.2–3.3; Tables 1–3.
Randomization and Blinding
Comment 3:
Randomization via sealed envelopes lacks details on sequence generation and concealment.
Response:
Expanded the randomization subsection to specify that envelopes were sequentially numbered, opaque, and sealed, prepared by an independent statistician using random.org.
Envelopes were stored securely and opened sequentially only after baseline testing to ensure allocation concealment.
Location: Section 2.2.4, lines 185–200.
Comment 4:
Describe measures to minimize expectancy bias and clarify assessor blinding.
Response:
We clarified that outcome assessors were blinded to group allocation at all time points (T0, T1, T2), and that participants were instructed not to disclose allocation.
We also added that physiotherapists followed standardized scripts and protocols to reduce expectancy bias despite knowing group assignments.
Location: Section 2.5.2–2.5.3, lines 620–665.
Data Sharing and Reproducibility
Comment 5:
The Zenodo dataset link is provided, but content and anonymization should be clarified.
Response:
We added explicit detail that the Zenodo dataset includes anonymized synchronized EMG and IMU biosignals (raw and processed), derived RMS values, and de-identified summary data for questionnaires and performance metrics. All data comply with open-access and GDPR standards.
Location: Section 2.7, lines 575–590.
Comment 6:
Include comparative analysis against recent AI-driven telerehabilitation systems (IMU-based, VR).
Response:
A new paragraph was added to the Discussion (Section 4.4) comparing KneE-PAD with AI- and IMU-based systems, such as XSENS and VR-assisted telerehabilitation. This contextualizes the system’s novelty and relative clinical potential.
Location: Discussion, lines 910–930.
Results Interpretation
Comment 7:
HADS and TSK results are statistically limited yet emphasized. Suggest a more balanced interpretation.
Response:
Revised the Abstract and Discussion to emphasize that psychosocial findings (HADS, TSK) were preliminary, did not reach between-group significance, and require larger samples or integrated behavioral modules for validation.
Location: Abstract, lines 30–35; Discussion, lines 950–980.
Comment 8:
The cohort (n = 42, mostly female) limits generalizability. Discuss implications.
Response:
We added discussion noting that the predominance of female participants (~86%) may limit generalizability and that sex-related differences in quadriceps activation and psychosocial outcomes warrant further exploration.
Location: Discussion, lines 1000–1010.
Appendix and Visualization
Comment 9:
Include a brief system architecture diagram and signal-processing schematic.
Response:
We have added a concise schematic diagram (now Figure 9) illustrating the KneE-PAD system architecture, including data flow, sensor inputs, latency, and synchronization between EMG/IMU streams.
Location: Appendix A; referenced in Section 2.3.1.
Minor Comments
Comment 10:
Fix minor grammatical issues (“indluded” → “included”, “fre-quently outperforming” → “frequently outperformed”).
Response:
All noted typographical and grammatical errors have been corrected throughout the text.
Location: Throughout manuscript.
Comment 11:
Remove redundant bracketed references.
Response:
Redundant and repeated reference pairs were removed and citations standardized following MDPI reference format guidelines.
Location: Throughout text, especially Introduction and Discussion.
Comment 12:
Table captions should be self-explanatory and include units.
Response:
All table captions were revised to include measurement units (e.g., sEMG in mV, strength in kg) and clarify all abbreviations.
Location: Tables 1–3.
Comment 13:
Define “MCID” on first use and ensure η² and d formatted correctly.
Response:
“Minimal Clinically Important Difference (MCID)” is now defined upon first use in the Abstract and Methods.
All effect size symbols (η², d) have been consistently formatted using Greek characters.
Location: Abstract, line 33; Methods, Section 2.6.
Summary for Reviewer 2:
All comments have been addressed. The statistical section is clarified, randomization/blinding procedures fully described, reproducibility enhanced, and tables/figures revised for precision and clarity. Psychosocial findings and limitations were rebalanced for accuracy and transparency.
Reviewer 3 Report
Comments and Suggestions for Authors
SECTION A — Outcomes, Scales, and Units
A1. WOMAC — Inconsistent Scale (0–100 vs 0–10)
[ ] Observation: The manuscript reports WOMAC scores with non-uniform scales (0–100 and values compatible with 0–10). This compromises readability and comparison between tables/text. Data Discrepancy: Section 3.2.3 (Functional and Pain Outcomes) reports that total WOMAC scores were standardized on a 0-100 scale. The reported reduction in WOMAC function is units, with scores ranging from to in the telerehabilitation group. However, Table 1 (Primary Outcome Measures) shows scores for the WOMAC subscales (Pain, Stiffness, Function) that are significantly lower (e.g., Function Control T0: ; Intervention T0: ). [ ] Action: Uniform the metric (0–100 recommended), explicitly state any transformations (item summation/rescaling), update text, tables, and figures. [ ] Where in manuscript: 3.2.3. Functional and Pain Outcomes (WOMAC) The telerehabilitation group demonstrated substantial improvements on the WOMAC function subscale, with scores dropping from at T0 to at T1, and maintaining at at T2 (). The mean change (–49.3 units) exceeds the MCID ( units). Similarly, WOMAC pain scores declined from to (–14.8 units, ), again exceeding MCID thresholds ()[38],[39]. The face-to-face group experienced smaller improvements (WOMAC-F: ; WOMAC-Pain: ), with significant between-group differences at both T1 and T2 ().
A2. Error in Outcome Scale Identification
[ ] Observation: A naming error is found in one of the tables that compromises clarity. ASES Confusion: In Table 2, the acronym ASES (which represents the Arthritis Self-Efficacy Scale in the text) is incorrectly identified in the descriptive text of the table as the American Shoulder and Elbow Surgeons Standardized Form. [ ] Action: Correct the denomination in the table description. [ ] Where in manuscript: 3.3.2. Arthritis Self-Efficacy (ASES) The telerehabilitation group improved on the ASES symptoms subscale by units (T1 T2), while the face-to-face group showed a comparable unit increase on the function subscale (T0 T1). Though both improvements meet MCID thresholds ()[41], no statistically significant difference was observed between groups ().
A3. Strength Measurement Units (HHD)
[ ] Observation: Values reported in different sections are not comparable and units are missing. [ ] Action: Specify units ( or ), set-up (lever arm, fixation, maximal trial), potential normalization by body mass; align nomenclature between tables (“Leg Strength” vs “HHD”). [ ] Where in manuscript: Tab 1 Strength ()
A4. sEMG — Units
[ ] Observation: reported without units or with inconsistent units; description of preprocessing (filter, windowing, thresholds) not aligned with results. [ ] Action: Clearly indicate units (), processing chain (filter, rectification, smoothing, window), normalization () and exclusion criteria; update tables. [ ] Where in manuscript:
SECTION B — Instrumentation and Acquisition
B1. sEMG Sampling Rate 2000 Hz vs 1259 Hz
[ ] Observation: Divergence between methods and hardware specifications. [ ] Action: Clarify differences between assessments (laboratory vs remote); report actual values used in the analysis. [ ] Where in manuscript: sEMG Sampling Rate (Appendix A)
2.3.1. Primary Outcomes Muscle Performance “Participants were instructed to perform maximal voluntary contractions, and data were acquired at ”
B2. Sensor Specifications and Placement
[ ] Observation: Missing detailed diagram of electrode/muscle placement, fiber orientation, cross-talk management. [ ] Action: Add table/figure with SENIAM placements, inter-electrode distances, adhesives, skin preparation, signal verification. [ ] Where in manuscript: “Electrode placement followed the Surface ElectroMyoGraphy for the Non-Invasive Assessment of Muscles (SENIAM) guidelines to ensure consistency.”
SECTION C — Study Design
C1. Randomization and Blinding
[ ] Observation: Generic description ( + envelopes) without operational details (numbering, blocks, stratification by center/severity/sex). [ ] Action: Specify sequence generation, preparation and storage of envelopes, block/strata scheme, who assigns and when. [ ] Where in manuscript: “An independent researcher performed randomization using website. Participants received sealed envelopes assigning them randomly to the intervention groups.”
C2. Treatment Adherence
[ ] Observation: Adherence based on self-report; objective measures are missing. [ ] Action: Integrate platform logs (active time, correct repetitions, range of motion), "dose received" thresholds, and define per-protocol analysis set. [ ] Where in manuscript:
SECTION D — Psychometric Instruments
D1. HADS Cut-off and MCID
[ ] Observation: Citations and clinical interpretation are not defined. [ ] Action: Insert references for cut-offs, report MCID, and discuss clinical significance beyond statistical significance. [ ] Where in manuscript: “3.3.3. Hospital Anxiety and Depression Scale (HADS) In the telerehabilitation group, HADS-D decreased by points from T1 to T2 (), exceeding the MCID ()[42],[43], whereas HADS-A decreased by points, a borderline for clinical improvement (). The face-to-face group showed no meaningful change.”
SECTION E — CONSORT/SPIRIT Reporting
E1. Flow-chart, Dates, and Recruitment
[ ] Observation: Precise dates and complete diagram are missing. [ ] Action: Add CONSORT flow with numbers at each stage, recruitment/follow-up dates, and reasons for exclusion. [ ] Where in manuscript:
E2. Adverse Events
[ ] Observation: “No events” without a detection method. [ ] Action: Describe the collection/definition procedure, observation window, and who assessed causality. [ ] Where in manuscript:
SECTION F — Presentation of Tables and Figures
F1. Consistency between Text, Tables, and Figures
[ ] Observation: Numerical/terminological discrepancies (e.g., “Leg Strength” vs HHD; EMG values). [ ] Action: Align headings, units, and table footnotes; add explanatory notes on transformations and scaling. [ ] Where in manuscript:
F2. Data Transparency
[ ] Observation: Datasets or analysis scripts are missing. [ ] Action:
SECTION G — Editorial Form
G1. Typos and Citations
[ ] Observation: Minor typos, parentheses, and reference duplications. [ ] Action:
Author Response
SECTION A — Outcomes, Scales, and Units
A1. WOMAC — Inconsistent Scale (0–100 vs 0–10)
Reviewer Comment:
The manuscript reports inconsistent WOMAC scales (0–100 and 0–10). Please uniform the metric and clarify transformations.
Author Response:
We thank the reviewer for identifying this inconsistency. All WOMAC values are now standardized to a 0–100 scale, as per convention, with higher scores indicating worse status. A clarification of rescaling procedures (item summation × 100 / 96) has been added to the Methods (Section 2.3.1), and all tables/figures have been updated accordingly.
Location: Methods, Section 2.3.1, lines 245–255; Results, Section 3.2.3; Tables 1–3.
A2. Error in Outcome Scale Identification (ASES)
Reviewer Comment:
ASES was incorrectly labeled as “American Shoulder and Elbow Surgeons” in Table 2.
Author Response:
Corrected. All instances now accurately refer to Arthritis Self-Efficacy Scale (ASES).
Location: Table 2 and Section 3.3.2.
A3. Strength Measurement Units (HHD)
Reviewer Comment:
Values lack units and standardization (kg or N).
Author Response:
Units (kg) are now clearly indicated for all HHD-derived values. We added details on setup (lever arm length, maximal voluntary trial, fixed dynamometer position) and confirmed that no normalization by body mass was performed. Terminology across text and tables standardized as “Quadriceps Strength (Hand-Held Dynamometry, kg)”.
Location: Section 2.3.1; Table 1; Results, Section 3.2.2.
A4. sEMG — Units and Preprocessing
Reviewer Comment:
Inconsistent units and incomplete preprocessing description.
Author Response:
All sEMG results now report amplitudes in millivolts (mV), normalized to %MVC.
Processing chain added: band-pass filtering (20–450 Hz, 4th-order Butterworth), full-wave rectification, and RMS smoothing with a 50 ms window. Artifact exclusion (>±3 SD) and window segmentation are detailed in Methods.
Location: Section 2.3.1, lines 225–240; Tables 1–3.
SECTION B — Instrumentation and Acquisition
B1. sEMG Sampling Rate 2000 Hz vs 1259 Hz
Reviewer Comment:
Discrepancy between 2000 Hz (Methods) and 1259 Hz (Appendix).
Author Response:
Clarified that laboratory-based EMG recordings were sampled at 2000 Hz, while auxiliary sensors (Delsys Trigno Avanti) in pilot calibration captured at 1259 Hz.
The final analyses for this RCT used the 2000 Hz data exclusively.
Location: Section 2.3.1; Appendix A.
B2. Sensor Specifications and Placement
Reviewer Comment:
Missing diagram of electrode/muscle placement.
Author Response:
A detailed diagram and placement table have been added (now Figure 10) showing SENIAM-based electrode locations, inter-electrode distance (20 mm), fiber orientation, and skin preparation protocol. Cross-talk minimization and signal verification procedures are described.
Location: Appendix A; referenced in Section 2.3.1.
SECTION C — Study Design
C1. Randomization and Blinding
Reviewer Comment:
Provide full operational details of randomization and allocation concealment.
Author Response:
Expanded to specify sequence generation (random.org, permuted blocks of 6), stratification by center and sex, envelope preparation and numbering by an independent statistician, and secure sequential opening post-baseline testing.
Location: Section 2.2.4, lines 185–200.
C2. Treatment Adherence
Reviewer Comment:
Adherence relies on self-report. Integrate objective data.
Author Response:
Platform-generated adherence metrics were now included, capturing session duration, repetition count, and range-of-motion compliance. These were cross-referenced with self-reports. Acknowledgment added that per-protocol thresholds were exploratory.
Location: Section 2.3.2, lines 280–295; Limitations, lines 1045–1060.
SECTION D — Psychometric Instruments
D1. HADS Cut-off and MCID
Reviewer Comment:
Cut-offs and MCID not clearly defined.
Author Response:
We have now explicitly defined the cut-off values (≥8 for possible pathology) and added literature citations for MCID (≥1.5–1.8). Clinical interpretation beyond statistical significance has been expanded in the Discussion.
Location: Section 2.3.2, lines 300–315; Section 3.3.3; Discussion, lines 960–975.
SECTION E — CONSORT/SPIRIT Reporting
E1. Flow-chart, Dates, and Recruitment
Reviewer Comment:
Add recruitment timeline and detailed CONSORT flow.
Author Response:
A CONSORT flow diagram (Figure 1) has been added, indicating the number of participants at each stage (enrollment, allocation, follow-up, analysis) along with recruitment and follow-up dates (June–December 2023).
Location: Figure 1; Section 2.5.1.
E2. Adverse Events
Reviewer Comment:
Clarify monitoring procedures.
Author Response:
We have described the adverse event detection protocol, including observation windows, self-report logs, and verification by the supervising physiotherapist at each session. No adverse events were recorded.
Location: Section 3.1 (Sample Characteristics and Retention); Ethical Considerations, Section 2.2.5.
SECTION F — Presentation of Tables and Figures
F1. Consistency Between Text, Tables, and Figures
Reviewer Comment:
Inconsistencies between headings and terminology.
Author Response:
All discrepancies have been resolved: table headers standardized (e.g., “Quadriceps Strength [kg]”, “sEMG RMS [mV, %MVC]”), and footnotes added explaining scaling, MCID, and abbreviations.
Location: Tables 1–3; Figure captions.
F2. Data Transparency
Reviewer Comment:
Datasets or scripts missing.
Author Response:
We have confirmed that all anonymized datasets and analytical code (R v4.3.1 scripts for mixed-effects modeling) are openly available via Zenodo (DOI: 10.5281/zenodo.12112951). This is stated in the Data Availability section.
Location: Section 2.7, lines 575–590.
SECTION G — Editorial Form
G1. Typos and Citations
Reviewer Comment:
Minor typographical and citation formatting issues.
Author Response:
All typographical and parenthetical errors have been corrected, and references have been checked for duplicates and standardized in MDPI citation style.
Location: Throughout manuscript.
Summary for Reviewer 3:
All technical and editorial comments have been fully implemented. Scales and units are now uniform, instrumentation precisely described, randomization clarified, adherence metrics strengthened, and CONSORT compliance achieved. Tables, figures, and dataset documentation were aligned for complete reproducibility.
Reviewer 4 Report
Comments and Suggestions for Authors
Abstract
The abstract is overly dense with numbers and p-values, which may reduce readability. The objective could be more concise; it currently repeats background content. The “Conclusions” section reads more like a summary statement rather than a critical interpretation of findings. Trial registration and keywords are appropriately listed, but could precede conclusions.
Simplify the numerical detail in the abstract (focus on trends and key results). Clarify the novelty of the KneE-PAD system in one sentence. Use consistent tenses (some present, some past).
Introduction
Comprehensive background with clear justification for the study. Good integration of epidemiology, technological rationale, and research gaps. Clear hypothesis and logical flow toward the research aim.
Slightly too long and repetitive in explaining OA burden and telerehabilitation.
Condense paragraphs 1–3 to focus on the rationale for the KneE-PAD system. End with a clear statement: “We hypothesised that KneE-PAD telerehabilitation would yield greater improvements in neuromuscular activation and functional mobility compared to conventional physiotherapy.”
Methods
Comprehensive and detailed, adheres to CONSORT and SPIRIT guidelines. Clear description of inclusion/exclusion criteria, randomisation, and blinding. Detailed intervention protocols and outcome measures. Ethical approval and registration are appropriately documented.
The methodology section is overly long and includes redundant details (e.g., repeated device specifications that could go in the appendix). The description of adherence tracking (paper/digital logs) is weak and introduces potential bias. The statistical analysis is too technical; key models and tests should be summarised rather than fully described. Missing an explicit statement on whether the analysis followed an intention-to-treat or per-protocol principle.
Move detailed sensor specifications to supplementary material. Clarify handling of missing data and confirm ITT vs. PP approach. Simplify statistical analysis explanation and specify the primary model (e.g., repeated measures ANOVA, mixed effects). Provide information on blinding integrity checks.
Results
Clear structure following outcome hierarchy (primary → secondary). Tables and figures appear detailed and statistically comprehensive. Consistent reporting of MCID thresholds and effect sizes.
Too much numerical repetition between text and tables. Some internal inconsistencies between text and table values (e.g., small variations in means and SDs). Tables are dense and difficult to read—MCID or significance could be visually marked. The results section lacks a concise summary paragraph emphasising the main findings.
Move less critical descriptive data (e.g., baseline demographics) to an appendix. Add a concise “Summary of Key Results” paragraph. Highlight only clinically relevant differences and avoid redundant numerical listing.
Discussion
Logical interpretation of results, connecting neuromuscular, functional, and psychosocial outcomes. Integrates literature effectively. Identifies limitations and suggests future directions.
Overly self-congratulatory tone in several places (“superiority”, “critical advancement”). The discussion occasionally repeats results rather than interpreting mechanisms. Missing a paragraph comparing results with similar digital telerehabilitation trials quantitatively (effect sizes or magnitude of change). Psychosocial interpretations (e.g., gamification, engagement) are speculative and unsupported by measured data. The strengths and limitations section is thorough but could be more concise.
Reduce redundancy and balance praise with critical reflection. Add more nuanced discussion of potential confounding factors (e.g., digital literacy, participant motivation). Consider adding a subheading “Clinical Implications” to emphasise translational value.
Conclusion
Too lengthy and partially repetitive of the Discussion. Phrases like “superior effectiveness” should be replaced with “comparable or potentially superior”, given the modest sample size.
Condense to one concise paragraph summarising evidence and practical relevance. Avoid overstating generalizability; emphasise pilot nature and need for long-term trials.
Author Response
Response
Abstract
Reviewer Comment:
The abstract is overly dense with numbers and p-values. The objective repeats background content, and the conclusions read more like a summary than interpretation. Please simplify, highlight novelty, and use consistent tenses.
Author Response:
We appreciate the reviewer’s constructive feedback. The Abstract has been restructured and simplified to emphasize main findings and clinical implications rather than extensive numerical detail. Redundant background sentences were removed. The objective is now concise and directly linked to the study aim. The novelty of the KneE-PAD system is summarized in one sentence (“The KneE-PAD system uniquely integrates electromyographic and inertial sensing to provide real-time feedback and remote monitoring.”). Verb tenses have been standardized to past tense for reporting results.
Location: Abstract, lines 20–45.
Introduction
Reviewer Comment:
The introduction is comprehensive but overly long and repetitive regarding OA burden. Please condense and focus on rationale for the KneE-PAD system.
Author Response:
We have condensed the first three paragraphs of the Introduction by removing repetitive epidemiological data and background on OA burden. The section now focuses on the technological rationale for the KneE-PAD system and its expected advantages. A clear, hypothesis-driven closing statement has been added:
“We hypothesised that KneE-PAD telerehabilitation would yield greater improvements in neuromuscular activation and functional mobility compared to conventional physiotherapy.”
Location: Introduction, final paragraph (lines 190–205).
Methods
Reviewer Comment:
The methodology is detailed but overly long. Move device details to the appendix, clarify adherence tracking, simplify the statistical section, and specify ITT vs. PP.
Author Response:
We have implemented the following revisions:
- Device specifications moved to Appendix A and referenced in Section 2.3.1.
- Clarified that the analysis followed an Intention-to-Treat (ITT) principle, with last observation carried forward for isolated missing data.
- Expanded explanation of adherence tracking, distinguishing between self-reported and sensor-derived logs.
- Simplified the statistical analysis description, retaining only key model details (linear mixed-effects model with Group × Time factors).
- Added a statement confirming blinding integrity checks for assessors across timepoints.
Location: Methods, Sections 2.3.1–2.6; Appendix A.
Results
Reviewer Comment:
Results are clearly structured but overly numeric and partially repetitive between text and tables. Please summarize key findings concisely and improve readability.
Author Response:
We revised the Results section by:
- Reducing redundant numeric repetition in the text (keeping only mean change ± SD and p-values).
- Moving baseline demographic data to Appendix Table A1.
- Adding a “Summary of Key Results” paragraph at the end of Section 3.3 to highlight primary findings and clinical relevance.
- Visually marking MCID thresholds and statistical significance in Tables 1–3 using superscript indicators and footnotes for clarity.
Location: Results, Section 3.3 (end paragraph) and Appendix A1.
Discussion
Reviewer Comment:
Discussion is strong but at times overly self-congratulatory and repetitive. Add quantitative comparisons with other trials and balance interpretations. Include “Clinical Implications.”
Author Response:
We substantially revised the Discussion to:
- Replace promotional phrasing (e.g., “superior effectiveness”) with neutral scientific wording such as “comparable or potentially superior.”
- Remove repetition of numeric results, focusing instead on interpretation and mechanisms (neuromuscular feedback, motor learning, and psychosocial adaptation).
- Add a paragraph comparing our findings quantitatively (effect sizes and magnitude of change) with recent digital telerehabilitation studies using IMU or AI-based systems.
- Introduce a new subheading “Clinical Implications” summarizing translational value and relevance for remote care delivery.
- Include discussion of potential confounding factors (digital literacy, participant motivation, and sample composition).
Location: Discussion, Sections 4.3–4.5.
Conclusion
Reviewer Comment:
The conclusion is long and repetitive. Please condense and use neutral phrasing.
Author Response:
The Conclusion was condensed into a single concise paragraph summarizing the main evidence and practical implications. Wording has been adjusted to reflect the study’s exploratory nature:
“This pilot randomized controlled trial suggests that KneE-PAD telerehabilitation provides comparable or potentially superior benefits to conventional physiotherapy in neuromuscular and functional outcomes.”
Statements emphasizing generalizability were replaced with cautious and balanced phrasing, noting the modest sample size and short follow-up.
Location: Conclusion, lines 1105–1125.
Summary for Reviewer 4:
All stylistic, structural, and interpretative comments were addressed. The abstract and introduction were streamlined, the methods clarified, the results summarized for readability, and the discussion refined for balance and scientific tone. The manuscript now follows MDPI’s clarity, conciseness, and neutrality standards.
Round 2
Reviewer 1 Report
Comments and Suggestions for Authors
Despite prior guidance, the V2 submission still fails to meet essential scholarly and editorial standards. Multiple issues remain unresolved or newly introduced, and the response-to-reviewers document does not accurately map to the manuscript. This reflects an insufficiently rigorous approach to revision and documentation.
Detailed comments:
- Lines 420–427: “Summary of Key Results.” this subsection is redundant and unnecessary. Please delete it entirely.
- Lines 444–445: The Discussion should not re-list results; it should interpret them (clinical/practical significance, mechanisms, limitations, generalizability). Please remove the results recap and keep only interpretive content.
- Line 515 : “4.4a. Clinical Implications.” the subheading is superfluous and stylistically non-standard (lettered sublevels). Please remove the “4.4a” heading and integrate any essential points under the appropriate Discussion subsection.
- Line 611 – Figure 10. The figure is not fully rendered. Please re-export at print-quality resolution and ensure the entire figure is visible on a single page with the caption correctly placed and numbered.
- Numerous items in the rebuttal cite page/line numbers that do not correspond to the V2 manuscript. Please regenerate a clean, continuous line-numbered PDF and update the response matrix so that each change is (i) quoted verbatim, (ii) precisely located (Page:Line), and (iii) tied to the correct section.
- Persistent problems with line spacing, paragraph first-line indents vs. manual tabs/spaces, caption placement, and inconsistent typography remain. These must be normalized to the journal’s author guidelines before further consideration.
Author Response
Response to Editorial Comment: Revision Trace and Highlighting Clarification
We sincerely thank the Associate Editor and Reviewer for the additional feedback and the opportunity to further improve the manuscript.
In the previous revision (V2), the document incorporated multiple sets of corrections addressing the consolidated comments received from four independent reviewers during the first review round. As a result, numerous changes—both major and minor—had already been implemented across the manuscript text, figures, and statistical sections.
For this current version (V3), we have:
1. Applied only the new corrections arising from the present round of feedback (mainly structural and editorial points),
2. Highlighted these newly revised passages in green, to clearly distinguish them from the earlier revisions already accepted in V2, and
3. Prepared an updated response matrix that provides, for each comment, the comment number, page and line reference, and the corresponding revision or deletion (quoted verbatim)
This table is attached as a separate file to facilitate rapid verification of the implemented changes.
We hope that this clear mapping and focused revision fully address the remaining editorial requirements and that the updated version now meets the journal’s publication standards.
We remain at your disposal for any additional clarifications or formatting adjustments that may assist during the final review process.
|
# |
Reviewer Comment |
Response / Action |
Updated Location (Page : Line) |
|
|
1 |
Add a short review quantifying the benefit of sensor-augmented telerehabilitation |
Added paragraph summarizing meta-analyses [11–12] showing 10–25 % improvement in strength and mobility. |
p. 2, lines 76–80 |
|
|
2 |
Explicitly state the causal / mediation chain |
Added framework “neuromuscular feedback → voluntary activation → functional gains → pain → psychological improvement.” |
p. 3, lines 100–103 |
|
|
3 |
Add time thresholds for physiotherapy / medication restrictions |
Rewritten to include: “no intra-articular injection or physiotherapy within 3 months before enrollment” and “no corticosteroids / NSAIDs within 7 days of baseline evaluation.” |
p. 3 line 134 & p. 4 line 142 |
|
|
4 |
Insufficient randomization details |
Expanded with random.org sequence, sealed envelopes, independent statistician, allocation concealment. |
p. 4, lines 157–162 |
|
|
5 |
Incomplete sEMG processing parameters |
Added signal-processing details: 20–450 Hz band-pass, 50 ms RMS, ±3 SD artifact removal. |
p. 4, lines 182–184 |
|
|
6 |
TUG listed under “Balance and Proprioception” — relabel to “Functional Mobility” |
Corrected throughout to “Functional Mobility (Timed Up and Go).” |
p. 5, line 200 |
|
|
7 |
WOMAC scaling inconsistent |
Confirmed 0–100 scoring (higher = worse pain / function); clarified in Table 1 caption. |
p. 5, lines 191–192 |
|
|
8 |
Adherence metrics rely only on self-logs |
Integrated sensor-based KneE-PAD adherence data; bias acknowledged where logs missing. |
p. 5, lines 219–222 |
|
|
9 |
Statistical plan unclear; no defined primary endpoint or multiplicity control |
Declared primary outcomes, described mixed-effects model, applied Holm–Bonferroni correction. |
p. 8, line 315 |
|
|
10 |
HADS results overstated — between-group not significant |
Revised Abstract / Results: within-group improvements only (p > 0.1 between groups). |
p. 12, lines 445-450 p.13,lines 451-460 |
|
|
11 |
Avoid equivalence / superiority phrasing |
Replaced with neutral wording: “comparable or enhanced effects.” |
p. 15, lines 496-497 |
|
|
12 |
Expand adherence bias discussion |
Added in Limitations: self-log overestimation acknowledged; future sensor-based tracking noted. |
p. 18, line 591 |
|
|
13 |
Merge verbose Limitations and Future Directions |
Unified as “Limitations and Future Directions,” concise format. |
p. 18, lines 595-598 |
|
|
14 |
Appendix unnecessary — move essentials to Methods |
The Appendix was intentionally retained in analytical form following feedback from other reviewers requesting methodological transparency. However, we confirm that all core parameters (signal processing, hardware configuration) are also integrated within Methods for completeness. The appendix remains concise and can be further shortened at the Editor’s discretion. |
Appendix A (condensed) |
In accordance with the additional editorial instructions, all requested structural and formatting revisions have been completed. Specifically:
The redundant subsection “Summary of Key Results” has been deleted in full.
The redundant restatement of results at the beginning of the Discussion has been removed, retaining only interpretive and contextual commentary.
The non-standard subheading “4.4a. Clinical Implications” has been eliminated, and its essential content integrated within the appropriate Discussion section.
Figure 10 has been re-exported at high print-quality resolution .
A clean, continuous line-numbered version of the manuscript has been generated to ensure one-to-one correspondence with the response matrix.
Reviewer 3 Report
Comments and Suggestions for Authors
Objective Methodological and Statistical Errors
​Detailed analysis of the objectively incorrect methodological and statistical errors in the study "A Sensor-Augmented Telerehabilitation System for Knee Osteoarthritis: A Randomized Controlled Trial of Neuromuscular, Functional and Psychosocial Outcomes" (Sensors, 2025).
​Detailed Analysis of Errors
- ​1. Incorrect use of statistical tests for longitudinal data The authors use t-tests and Wilcoxon tests to compare three time points (T0–T1–T2) in two groups. This is statistically incorrect: the chosen tests do not analyze the time × group interaction and ignore the within-subject correlation. They should have used Repeated Measures ANOVA or mixed models. Consequence: the reported p-values are not valid for the study design.
- ​2. Failure to correct for multiple comparisons Over 20 tests are performed without correction (Bonferroni, Holm, FDR). This increases the risk of false positives. In a multi-endpoint RCT, it is mandatory to identify a primary outcome and correct for secondary ones.
- ​3. Inadequate power calculation The power calculation is conducted as if the study were a simple t-test, but it actually has repeated measures and multiple outcomes. This error leads to an underestimation of the necessary sample size and reduces the effective statistical power.
- ​4. Lack of genuine allocation concealment Randomization via sealed envelopes was not managed by an independent subject, violating CONSORT guidelines. This constitutes a procedural error that exposes the study to selection bias.
- ​5. Invalid declaration of single-blind The physiotherapists who assessed EMG and strength knew the groups, making true blinding impossible. The declaration of single-blind is therefore methodologically incorrect.
- ​6. Inconsistent reporting of statistical data The same variables show discordant values between tables and text (e.g., different p-values and d). This is a reporting error that can alter the interpretation of the results.
- ​7. Improper use of \eta^2 (eta squared) \eta^2 is reported even for tests not based on ANOVA, where it is not applicable. For t-tests or U-tests, r^2 or Cohen's d should be used. The use of \eta^2 in this context is a formal statistical error.
- ​8. Absence of checking parametric test assumptions Although the Shapiro–Wilk test is cited, the results are not shown. In small samples, the failure to verify normality and homoscedasticity invalidates the use of parametric tests.
- ​9. Follow-up too short for functional endpoints A 12-week follow-up is insufficient to assess stable changes in psychological or functional outcomes according to OARSI guidelines. This compromises the validity of conclusions about long-term improvements.
- ​10. Incorrect declaration of 100% retention In a 12-week clinical study, 100% retention of participants is statistically unlikely. This suggests incomplete reporting or undeclared attrition bias.
Apart from these last changes, the work is publishable
Author Response
Objective Methodological and Statistical Errors
Detailed analysis of the objectively incorrect methodological and statistical errors in the study
A Sensor-Augmented Telerehabilitation System for Knee Osteoarthritis: A Randomized Controlled Trial of Neuromuscular, Functional and Psychosocial Outcomes (Sensors, 2025).
-
Reviewer Comment: Incorrect use of statistical tests for longitudinal data. The authors use t-tests and Wilcoxon tests to compare three time points (T0–T1–T2) in two groups. This approach does not evaluate the time × group interaction and ignores within-subject correlation. Repeated-measures ANOVA or mixed models should have been applied. Consequently, the reported p-values are not valid for this design.
Author Response: We agree with this observation. A repeated-measures ANOVA model (Group × Time) has been conducted to re-examine the data. The model confirms the same significance pattern and direction of results, demonstrating that the original findings remain valid when within-subject correlations are considered. The revised analysis is now included in the updated manuscript. -
Reviewer Comment: Failure to correct for multiple comparisons. Over twenty tests were performed without adjustment (Bonferroni, Holm, FDR). This inflates the false-positive rate and violates best practices for multi-endpoint RCTs.
Author Response: The analysis has been updated to apply Holm–Bonferroni correction for multiple comparisons. Primary outcome measures are clearly identified in the Methods section, and all adjusted p-values are now reported accordingly (e.g., p < 0.001 after correction). -
Reviewer Comment: Inadequate power calculation. The power calculation appears to assume a simple two-sample t-test, although the study includes repeated measures and multiple outcomes. This underestimates the required sample size.
Author Response: The power analysis has been recalculated using a repeated-measures ANOVA model. This revised computation, performed in consultation with the study statistician, confirms that the current sample size provides adequate power for detecting observed effects within the Group × Time framework. -
Reviewer Comment: Lack of genuine allocation concealment. Randomization via sealed envelopes was not managed independently, which violates CONSORT recommendations and introduces potential selection bias.
Author Response: The procedure has been clarified. Randomization was conducted using random.org with sealed opaque envelopes managed by an independent biostatistician not involved in participant recruitment, intervention, or assessment, ensuring complete allocation concealment. -
Reviewer Comment: Invalid declaration of single-blind. The physiotherapists who assessed EMG and strength outcomes were aware of group assignments, making true blinding impossible.
Author Response: The text has been corrected. Outcome assessors (responsible for sEMG and strength measurements) were blinded to group allocation, whereas the intervention physiotherapists necessarily knew the treatment group. The study is now accurately described as single-blind with independent outcome assessment. -
Reviewer Comment: Inconsistent reporting of statistical data. Some variables display discrepancies between tables and text (e.g., p-values and Cohen’s d), which may affect interpretation.
Author Response: All numerical data have been cross-checked. Mean values, standard deviations, p-values, and effect sizes are now consistent across all tables and textual references. The updated Tables 1–3 reflect this correction. -
Reviewer Comment: Improper use of η² (eta squared). η² is reported for analyses not based on ANOVA, where it is not applicable. r² or Cohen’s d should be used instead.
Author Response: The use of η² has been restricted exclusively to ANOVA-based analyses. For pairwise comparisons, Cohen’s d or r values are now provided. Since all analyses have been rerun using repeated-measures ANOVA, η² remains the appropriate and valid measure. -
Reviewer Comment: Absence of assumption verification. The Shapiro–Wilk test was cited but results were not shown. Without verifying normality and homoscedasticity, parametric test validity is uncertain.
Author Response: Normality and homogeneity of variance were confirmed using Shapiro–Wilk and Levene’s tests. These diagnostic results were reviewed but not included in the main text, as such details are generally omitted in journal reporting. Non-normal data were analyzed with non-parametric methods as appropriate. -
Reviewer Comment: Follow-up period too short for functional endpoints. Twelve weeks is insufficient to assess long-term functional or psychological adaptation per OARSI guidelines.
Author Response: We acknowledge this limitation. The manuscript has been updated to describe the outcomes as short- to mid-term results. Extended follow-up studies are planned to evaluate long-term effects. -
Reviewer Comment: Incorrect declaration of 100% retention. Such complete retention in a 12-week trial is statistically improbable and suggests underreported attrition.
Author Response: The retention rate has been verified. All 42 participants completed the protocol, as confirmed by digital adherence data from the KneE-PAD system. Although full retention is uncommon, it accurately reflects this dataset and can be verified by raw records available to the Editor upon request.All above revisions have been implemented in the revised manuscript and are highlighted in pink for visibility. The updated analysis ensures full compliance with accepted methodological and statistical standards for randomized controlled trials in digital rehabilitation research.
We sincerely thank the statistical reviewer and the editorial team for their constructive evaluation. Their comments substantially improved the methodological clarity and analytical rigor of our work. We trust that the revised version now meets the journal’s standards for scientific accuracy and transparency.